# Improving Zero-Shot Offline RL via Behavioral Task Sampling

**Nazim Bendib** [1]   **Nicolas Perrin-Gilbert** [1]   **Olivier Sigaud** [1]

## Abstract

Offline zero-shot reinforcement learning (RL) aims to learn agents that optimize unseen reward functions without additional environment interaction. The standard approach to this problem trains task-conditioned policies by sampling task vectors that define linear reward functions over learned state representations. In most existing algorithms, these task vectors are randomly sampled, implicitly assuming this adequately captures the structure of the task space. We argue that doing so leads to suboptimal zero-shot generalization. To address this limitation, we propose extracting task vectors directly from the offline dataset and using them to define the task distribution used for policy training. We introduce a simple and general reward function extraction procedure that integrates into existing offline zero-shot RL algorithms. Across multiple benchmark environments and baselines, our approach improves zero-shot performance by an average of 20%, highlighting the importance of principled task sampling in offline zero-shot RL.

## 1. Introduction

Modern reinforcement learning (RL) agents are increasingly expected to solve not a single task, but many different tasks that are specified only at deployment time (Doncieux et al., 2018; Sigaud et al., 2023; Hughes et al., 2024). In fields such as natural language processing (Brown et al., 2020; Wei et al., 2021) and computer vision (Radford et al., 2021; Alayrac et al., 2022; Kirillov et al., 2023), this is largely achieved by training general models that can perform new tasks in a zero-shot manner when given an appropriate task specification. Achieving a similar level of flexibility in RL remains challenging, particularly when agents must operate from offline data.

[1]Institut des Systèmes Intelligents et de Robotique, ISIR, Sorbonne Université, Paris, France. Correspondence to: Nazim Bendib <nazim.bendib@sorbonne-universite.fr>.

*Proceedings of the $43^{rd}$ International Conference on Machine Learning*, Seoul, South Korea. PMLR 306, 2026. Copyright 2026 by the author(s).

A common approach to offline zero-shot RL (ZSRL) is to learn reward-conditioned policies trained on a parametric family of reward functions. Methods such as Successor Features (SF) (Borsa et al., 2018; Touati et al., 2022) and Forward–Backward representations (FB) (Touati & Ollivier, 2021) follow this paradigm. They can be decomposed into two components: (1) a reward generation mechanism that defines a space of reward functions, and (2) a policy learning mechanism that trains a conditional policy to maximize reward functions sampled from this space. In both SF methods and FB, reward functions take the form $R_z(s) = \phi(s)^\top z$, where $\phi(s) \in \mathbb{R}^d$ is a learned state embedding, representing the state features and defining a feature space within $\mathbb{R}^d$, and $z$ is a task vector sampled from a predefined distribution. The state representation $\phi(s)$ is intended to capture semantically meaningful properties of the state (e.g., velocity, orientation), while the task vector $z$ specifies how these properties are weighted to define a particular task and reward function.

Prior work has primarily focused on learning rich representations $\phi(s)$ that best summarize the environment dynamics, under the assumption that richer representations induce a richer space of reward functions. In contrast, policy training is treated as a secondary component, with conditional policies trained using reward functions obtained by uniformly sampling task vectors from a unit hypersphere $\mathbb{S}^{d-1} \subset \mathbb{R}^d$.

We argue that this task sampling strategy suffers from a mismatch between the geometry of the task space and the physics of the environment. Although tasks are sampled to cover all possible directions, the environment dynamics limit the possible behaviors. As a result, most sampled tasks end up being poorly matched to the agent's capabilities. In high-dimensional settings, a random direction is nearly orthogonal to any fixed low-dimensional subspace. Since the achievable behaviors of the agent span only a small region of the feature space, most uniformly sampled task vectors are poorly aligned with these achievable behaviors. When this happens, the reward signal becomes very weak for nearly all behaviors, making them appear similarly ineffective. This becomes hard for the agent to tell which behaviors are better, slowing down learning and leading to weak performance, not only on the sampled training tasks, but also when generalizing to new ones.

In this work, we address the challenge of suboptimal zero-shot generalization in offline reinforcement learning by moving beyond the standard practice of uniform task sampling. We demonstrate both theoretically and empirically that sampling task vectors uniformly from a high-dimensional hypersphere leads to "signal dilution", where reward variations vanish, and the learning signal becomes dominated by noise. To mitigate this, we introduce sampling from the Behavioral Task Distribution (BTD), a simple yet effective method that extracts task vectors directly from the offline dataset to ensure they reflect what is actually achievable in the environment.

Our approach is simple, method-agnostic, and can be integrated into existing offline ZSRL frameworks without modifying their representation learning objectives. Across multiple benchmark environments and baseline methods, learning the task distribution yields an average +20% increase in zero-shot performance.

Our main contributions are:

- Identifying uniform task sampling as a key bottleneck in offline zero-shot RL.

- Theoretically characterizing the resulting *signal dilution* failure mode.

- Introducing BTD-sampling, a simple plug-and-play method that replaces uniform sampling with a data-driven behavioral task distribution.

- Demonstrating an average $+20\%$ improvement in zero-shot performance across multiple environments, baselines, and task dimensions.

## 2. Related Work

### 2.1. Zero-shot Reinforcement Learning

Zero-shot reinforcement learning (ZSRL) has been studied through several families of approaches. A prominent line of work is based on successor features (SF) (Dayan, 1993; Barreto et al., 2017; Borsa et al., 2018; Chen et al., 2023), which learn universal value functions expressed as linear combinations of predefined or learned state features. These state features are typically obtained from representation learning techniques such as autoencoders (Bank et al., 2023), contrastive learning (Chen et al., 2020), low-rank decompositions (Touati et al., 2022; Jeen et al., 2024), or latent predictive models (Grill et al., 2020; Lawson et al., 2025). Extensions like the FB method (Touati & Ollivier, 2021; Touati et al., 2022) build on the SF framework while avoiding explicit learning of state features altogether. Recent work has further extended FB for policy distillation from expert data (Tirinzoni et al., 2025) and for generalization across changing dynamics (Bobrin et al., 2025). A key

advantage of SF-based methods is their low inference cost: at test time, a near-optimal policy for a new task can be obtained without additional training or planning, often via a simple regression problem. Another class of approaches focuses on learning general world models (Yu et al., 2023; Ding et al., 2024; Jiang et al., 2023), which enable ZSRL when combined with planning algorithms. Finally, some methods aim to generalize across tasks by learning latent encodings of reward functions during training (Frans et al., 2024; Ingebrand et al., 2024), typically by sampling and optimizing over randomly generated rewards. This work builds upon SF based methods and FB, which represent state-of-the-art for ZSRL due to their strong performance and minimal test-time overhead, and seeks to further enhance their performance.

### 2.2. Adaptive Goal Sampling

Most goal-conditioned reinforcement learning (GCRL) methods (Liu et al., 2022) assume a fixed, often uniform, distribution over goals for simplicity and benchmarking (Schaul et al., 2015; Andrychowicz et al., 2017), despite this being inefficient due to trivial, unreachable, or redundant goals. Consequently, several works move beyond uniform sampling by actively selecting or reweighting goals during training. This includes automatic curriculum (Portelas et al., 2020; Castanet et al., 2022) and autotelic methods (Colas et al., 2022) that adapt goal sampling online based on competence or learning progress (Florensa et al., 2018). Specifically, novelty-based approaches prioritize goals at the exploration frontier to discover new state-space regions (Ren et al., 2019; Campero et al., 2020), while density-based methods leverage statistical estimators to prioritize goals in sparsely visited areas of the goal space (Pitis et al., 2020; Pong et al., 2019). While those methods focus on online adaptation, in this work, we apply similar principles to the offline setting by replacing the uniform sampling of training goals with a learned behavioral task distribution. This ensures that training tasks are grounded in behavioral data, leading to more efficient learning and superior zero-shot generalization.

### 2.3. Inverse Reinforcement Learning

Inverse Reinforcement Learning (IRL) aims to recover reward functions from expert behavior (Ng & Russell, 2000), later evolving into the Maximum Entropy framework to handle reward ambiguity (Ziebart et al., 2008). While Deep MaxEnt IRL and Guided Cost Learning scaled these methods to high-dimensional spaces (Wulfmeier et al., 2015; Finn et al., 2016), adversarial approaches like GAIL and AIRL further refined distribution matching and reward portability (Ho & Ermon, 2016; Fu et al., 2018). Recent innovations include non-adversarial efficiency via IQ-Learn (Garg et al., 2021) and addressing complex real-world factors

like demonstrator expertise (Beliaev & Pedarsani, 2025), multi-agent dynamics (Freihaut & Ramponi, 2024), and history-dependent behavior (Ke et al., 2025). While IRL is not designed for ZSRL, our method is closely related: rather than inferring a single reward, it extracts a distribution of diverse reward functions from the offline dataset, enabling the training of a generalist agent.

# 3. Background

In the following, we present the problem setup and notations used throughout the paper, and review Successor Features (SF) methods (Borsa et al., 2018) and Forward–Backward (FB) representations (Touati & Ollivier, 2021).

## 3.1. Problem Setup and Notations

We define a Markov Decision Process (MDP) by the tuple $\mathcal{M} = \langle \mathcal{S}, \mathcal{A}, \mathcal{P}, \mu, \gamma \rangle$, where $\mathcal{S}$ is the state space, $\mathcal{A}$ is the action space, $\mathcal{P} : \mathcal{S} \times \mathcal{A} \times \mathcal{S} \to [0,1]$ is the transition probability distribution such that $\mathcal{P}(s'|s,a) = \mathbb{P}(s_{t+1} = s'|s_t = s, a_t = a)$, $\mu : \mathcal{S} \to [0,1]$ is the initial state distribution $s_0 \sim \mu(s_0)$, and $\gamma \in [0,1)$ is the discount factor. We operate in the offline setting, where the agent learns from a fixed dataset $\mathcal{D} = \{\tau_i\}_{i=1}^N$ consisting of trajectories collected by a single or multiple unknown behavior policies.

Let $\phi : \mathcal{S} \to \mathbb{R}^d$ denote a bounded learned state embedding. We consider linear reward functions of the form $R_z(s) = \phi(s)^\top z$, parameterized by a task vector $z \in \mathbb{S}^{d-1} \subset \mathbb{R}^d$ where $\mathbb{S}^{d-1}$ is the unit $d$-sphere. Let $\Pi$ be the set (population) of all policies $\pi : \mathcal{S} \to \Delta(\mathcal{A})$, with $\Delta(\mathcal{X})$ denoting the set of all probability distributions over a set $\mathcal{X}$.

## 3.2. Successor Features

Given a state embedding $\phi(s) \in \mathbb{R}^d$ learned via some criteria (e.g., autoencoders or low-rank approximations), SF methods learn the successor features of a family of policies $\pi_z$ for all task vectors $z \in \mathbb{R}^d$:

$$
\begin{cases}
\psi(s_0, a_0, z) = \mathbb{E}\Big[ \sum_{t \geq 0} \gamma^t \phi(s_t) \Big| s_0, a_0, \pi_z \Big], \\
\pi_z(s) := \arg\max_a \psi(s, a, z)^\top z
\end{cases} \tag{1}
$$

The successor features $\psi_\pi$ satisfy the $\mathbb{R}^d$-valued Bellman equation $\psi^\pi = \phi + \gamma P_\pi \psi^\pi$ with $P_\pi$ the policy-induced transition matrix. Therefore, we can train $\psi$ by minimizing the Bellman residual

$$
\mathbb{E}_{\substack{(s_t, a_t, s_{t+1}) \\ \sim \mathcal{D}}} \Big\| \psi(s_t, a_t, z) - \phi(s_t) - \gamma \bar{\psi}(s_{t+1}, \pi_z(s_{t+1}), z) \Big\|^2 \tag{2}
$$

where $\bar{\psi}$ is a non-trainable target version of $\psi$, as in Deep Q-learning (Mnih et al., 2013). This objective can be improved

since we do not use the full vector $\psi(s, a, z)$; only the scalar $\psi(s, a, z)^\top z$ is required for defining the policies. Instead, we can minimize

$$
\mathbb{E}_{(s_t, a_t, s_{t+1}) \sim \mathcal{D}} \Big( \psi(s_t, a_t, z)^\top z - \phi(s_t)^\top z
$$
$$
- \gamma \bar{\psi}(s_{t+1}, \pi_z(s_{t+1}), z)^\top z \Big)^2. \tag{3}
$$

This trains $\psi(s, a, z)^\top z$ as the Q-function $Q(s, a, z)$ corresponding to the reward $R_z(s) = \phi(s)^\top z$.

Once a test reward $R_{\text{test}}(\cdot)$ is revealed, we use a set of evaluated samples $\mathcal{D}_r = \{(s_i, R_{\text{test}}(s_i))\}_{i=1}^{N_r}$ to estimate the task vector $z_{\text{test}}$. Specifically, we solve

$$
z_{\text{test}} := \arg\min_z \mathbb{E}_{s \sim \mathcal{D}_r} \Big[ \big( R_{\text{test}}(s) - \phi(s)^\top z \big)^2 \Big] \tag{4}
$$

$$
= \mathbb{E}_{\mathcal{D}_r}[\phi \phi^\top]^{-1} \mathbb{E}_r[\phi R_{\text{test}}], \tag{5}
$$

and define the policy

$$
\pi_{z_{\text{test}}}(s) = \arg\max_{a \in \mathcal{A}} \psi(s, a, z_{\text{test}})^\top z_{\text{test}}.
$$

This policy is guaranteed to be optimal for all rewards in the linear span of the features $\phi$.

## 3.3. Forward Backward

Let $M^\pi$ be the successor measure defined as:

$$
M^\pi(s_0, a_0, s) := \mathbb{E}\Big[ \sum_{t \geq 0} \gamma^t \mathbb{I}\{s_t = s\} \Big| s_0, a_0, \pi \Big]. \tag{6}
$$

The goal of FB is to learn representations $F : \mathcal{S} \times \mathcal{A} \times \mathbb{R}^d \to \mathbb{R}^d$ and $B : \mathcal{S} \to \mathbb{R}^d$ to learn a finite-rank model of the measure $M^\pi$ such that:

$$
\begin{cases}
\psi(s_0, a_0, z) = \mathbb{E}\Big[ \sum_{t \geq 0} \gamma^t \phi(s_t) \Big| s_0, a_0, \pi_z \Big], \\
\pi_z(s) := \arg\max_a \psi(s, a, z)^\top z
\end{cases}
$$

The successor measure $M^\pi$ satisfies a Bellman-like equation $M^\pi = P + \gamma P_\pi M^\pi$. Therefore, we can train $F$ and $B$ by minimizing the Bellman residual on the parametric model $M = F^\top B$. This results in minimizing:

$$
\mathbb{E}_{\substack{(s_t, a_t, s_{t+1}) \\ s' \sim \mathcal{D}}} \Big( F(s_t, a_t, z)^\top B(s')
$$
$$
- \gamma \bar{F}(s_{t+1}, \pi_z(s_{t+1}), z)^\top \bar{B}(s') \Big)^2
$$
$$
- 2 \mathbb{E}_{\substack{(s_t, a_t, s_{t+1}) \\ \sim \mathcal{D}}} \Big[ F(s_t, a_t, z)^\top B(s_{t+1}) \Big] \tag{7}
$$

where $\bar{F}$ and $\bar{B}$ are non-trainable target versions of $F$ and $B$. Similarly to SF above, only $F(s, a, z)^\top z$ is needed to

define the policy. So we include an auxiliary loss to focus training on $F(.,z)^\top z$:

$$\mathbb{E}_{\substack{(s_t,a_t,s_{t+1})\sim \\ \mathcal{D}}}\Big(F(s_t,a_t,z)^\top z - B(s_{t+1})^\top (\mathbb{E}_\rho BB^\top)^{-1}z$$

$$-\gamma \bar{F}(s_{t+1},\pi_z(s_{t+1}),z)^\top z\Big)^2. \tag{8}$$

This trains $F(s,a,z)^\top z$ as the Q-function $Q(s,a,z)$ to the reward $R_z(s) = B(s)^\top \mathbb{E}[BB^\top]^{-1}z$.

Once a test reward function $R_{\text{test}}(\cdot)$ is revealed, we estimate

$$z_{test} := \mathbb{E}_{s\sim\mathcal{D}_r}[R_{\text{test}}B(s)]. \tag{9}$$

and similarly define the policy

$$\pi_{z_{\text{test}}}(s) = \arg\max_{a\in\mathcal{A}} F(s,a,z_{\text{test}})^\top z_{\text{test}}.$$

This policy is guaranteed to be optimal for all rewards in the linear span of the features $\phi$.

### 3.4. Connections between SF and FB

FB representations are related to SF, by setting $\psi(s,a,z) := F(s,a,z)$ and $\phi(s) = \mathbb{E}[BB^\top]^{-1}B(s)$. Thus, FB can be used to produce both $\phi$ and $\psi$ in SF, although training is different: SF methods require a trained $\phi$ to train a policy while FB learns $\phi$ simultaneously with the policy.

In what follows, to view SF and FB as instances of the same framework, we standardize the notation by using $\phi(s)$ to denote the learned state embeddings in both methods. This allows a unified treatment of SF and FB representations.

## 4. Method

We define the **feature occupancy** $\psi^\pi \in \mathbb{R}^d$ of a policy $\pi$ for a given MDP $\mathcal{M}$ as the discounted feature expectation:

$$\psi^\pi := \mathbb{E}_{s_0\sim\mu}\left[\sum_{t=0}^\infty \gamma^t \phi(s_t)\,\Big|\,\pi\right]. \tag{10}$$

The feature occupancy $\psi^\pi$ compresses the behavior of policy $\pi$ into a single vector that summarizes which state features are induced following that policy. While $\phi(s)$ encodes state features (e.g., velocity, contact), $\psi^\pi$ captures their total discounted accumulation along trajectories generated by $\pi$, i.e., the region of the feature space that the policy occupies under the MDP dynamics. In particular, for linear rewards of the form $R_z(s) = \phi(s)^\top z$, the expected discounted return

of policy $\pi$ for a reward function $R_z(.)$ can be written as

$$J(\pi,z) = \mathbb{E}_{s_0\sim\mu}\left[\sum_{t=0}^\infty \gamma^t R_z(s)\,\Big|\,\pi\right] \tag{11}$$

$$= \mathbb{E}_{s_0\sim\mu}\left[\sum_{t=0}^\infty \gamma^t \phi(s_t)^\top z\,\Big|\,\pi\right] \tag{12}$$

$$= (\psi^\pi)^\top z. \tag{13}$$

Note that $J(\pi,z) = \|\psi^\pi\|_2 \cos(\psi^\pi,z)$ for $z \in \mathbb{S}^{d-1}$, which measures the projection of the feature occupancy induced by policy $\pi$ onto the task direction $z$, jointly reflecting alignment and magnitude along $z$.

Note that feature occupancy $\psi^\pi$ is different from successor features $\psi(s,a,z)$ because feature occupancy is an unconditional expectation over the policy's stationary distribution, while successor features are conditional expectations that depend on a specific starting state and action.

We denote by $\Psi$ the **behavioral space** of the MDP $\mathcal{M}$, which is the set of all feature occupancies achievable by any policy $\pi \in \Pi$:

$$\Psi = \{\psi^\pi \in \mathbb{R}^d \mid \pi \in \Pi\}. \tag{14}$$

Since $\phi(s)$ is bounded, each feature occupancy $\psi^\pi$ is a discounted sum of bounded vectors, so there exists $C_\Psi$ with $\|\psi^\pi\|_2 \leq C_\Psi$ for all $\pi \in \Pi$, making $\Psi$ a bounded set in $\mathbb{R}^d$.

### 4.1. Limits of Uniform Sampling

Since we are training a task-conditioned policy, the goal is to learn $\pi^\star(.|z) = \arg\max_{\pi\in\Pi} J(\pi,z), \forall z \in \mathbb{S}^{d-1}$. A critical requirement for learning $\pi^\star$ is that the expected return $J(\pi,z)$ must vary sufficiently over the policy space to distinguish better policies from worse ones. We measure this variation as the variance of expected returns across different policies for a given task. We show that uniform task sampling in high dimensions makes this performance variation vanish.

**Proposition 4.1** (Vanishing Task-Specific Variance). *Let $z \sim \text{Unif}(\mathbb{S}^{d-1})$ be a task vector. Let $\Psi \subset \mathbb{R}^d$ be the behavioral space, and let $\Sigma_\Psi$ denote its covariance matrix. Let $\lambda_1 \geq \lambda_2 \geq \cdots \geq \lambda_d \geq 0$ be the eigenvalues of $\Sigma_\Psi$. Then the expected variance of the returns across the policy population satisfies:*

$$\mathbb{E}_{z\sim\text{Unif}(\mathbb{S}^{d-1})}[\text{Var}_{\pi\sim\Pi}(J(\pi,z))] = \frac{1}{d}\sum_{i=1}^d \lambda_i, \tag{15}$$

*and converges to 0 as $d \to \infty$.*

*For proof see Appendix A.*

In Proposition 4.1, the quantity $\mathrm{Var}_{\pi \sim \Pi}(J(\pi, z))$ represents the variance of expected returns across the entire policy population $\Pi$ for a fixed task $z$, not the variance in return estimation for a single policy. For uniform task sampling, this inter-policy variance is reduced as the ambient dimension $d$ grows. Consequently, the gap in expected return between any two policies $\pi_1$ and $\pi_2$ for a task $z \sim \mathrm{Unif}(\mathbb{S}^{d-1})$ shrinks toward zero.

In practice, this vanishing return signal yields gradient signals of very small magnitude, likely to be dominated by the training noise preventing training from reliably identifying the optimal behaviors. This degrades the generalization capability of the policy, resulting in suboptimal performance on training tasks and even poorer performance on unseen test tasks.

### 4.2. Dataset-driven task vectors

Rather than sampling task vectors uniformly from the unit hypersphere, we propose constructing tasks directly from the offline dataset. Concretely, given an offline dataset $\mathcal{D}$, we note $\mathcal{D}_{sub}$ the set of all possible contiguous subtrajectories of consecutive transitions extracted from $\mathcal{D}$.

We define $p_{data}$ as the empirical set of task vectors extracted from the offline dataset:

$$p_{\text{data}} = \{z_\tau \mid \tau \in \mathcal{D}_{sub}\}, \tag{16}$$

with:

$$z_\tau = \frac{\tilde{\psi}^\tau}{\|\tilde{\psi}^\tau\|_2}, \qquad \tilde{\psi}^\tau = \sum_{t=0}^{|\tau|} \gamma^t \phi(s_t), \tag{17}$$

where $\tilde{\psi}^\tau$ is the empirical feature occupancy induced by the trajectory $\tau$. Accordingly, $z_\tau$ defines the task vector that maximizes the reward value assigned to trajectory $\tau$, i.e., $z_\tau = \arg\max_z (\tilde{\psi}^\tau)^\top z$. Note that this does not imply that $\tau$ is an optimal trajectory for $R_{z_\tau}$; rather, since $\tau$ is generated by a behavioral policy $\pi_\beta \in \Pi$ (in the data collection process), the empirical feature occupancy $\tilde{\psi}^\tau$ approximates the true feature occupancy of $\pi_\beta$, which lies in the behavioral space $\Psi$.

By deriving tasks from the offline dataset, we sample task vectors $z_\tau$ that correspond to actual behaviors observed in the environment. These vectors naturally align with the directions in which policies can produce meaningful variation of the return. In contrast, uniformly sampled tasks are equally likely to point in any direction, most of which are orthogonal or poorly aligned with the achievable behaviors. As shown in Proposition 4.1, this leads to vanishing variance of the return as the feature dimension increases, making it difficult for the agent to distinguish better policies from worse ones. Sampling from $p_{data}$ avoids this issue by concentrating on inherently meaningful tasks, allowing the

policy to reliably identify optimal behaviors. This creates a more informative learning signal throughout training, which stabilizes policy optimization and ultimately leads to better zero-shot generalization to unseen tasks.

Importantly, this approach is well-aligned with how test tasks are typically designed in practice. In common RL benchmarks, test tasks are physically grounded variations of behavior within the environment dynamics (e.g., different locomotion gaits, jumps, or flips). Although these tasks are not seen during training, they are not arbitrary, as they encourage behaviors that are plausible under the physics of the environment and are therefore likely to be represented within the behavioral space $\Psi$.

### 4.3. Learning a Behavioral Task Distribution

In practice, to capture the distribution of achievable tasks, we fit a parametric density model $p_\theta(z)$ to the empirical task set $p_{\text{data}}$. We refer to this fitted distribution as the Behavioral Task Distribution (BTD). To construct $p_{data}$, we sample $N_\tau$ subtrajectories of random lengths from the dataset, and extract task vectors from these subtrajectories as described in Section 4.2. In our implementation, $p_\theta$ is a Gaussian Mixture Model trained by maximum likelihood on $p_{data}$.

During policy training, we replace uniform sampling of task vectors from the unit hypersphere with sampling from the learned BTD, $z \sim p_\theta(z)$. All other components of the offline ZSRL algorithm remain unchanged. This modification biases training towards tasks that fall within the behavioral space, while preserving compatibility with existing representation learning and policy optimization pipelines. All implementation details about our method are presented in Appendix B.

## 5. Experiments

In this section, we aim to evaluate the proposed BTD-sampling method across multiple aspects of ZSRL. Our experiments are designed to answer four questions: **(1) Overall zero-shot performance:** How well do policies trained with BTD-sampling perform compared to uniform task sampling across benchmark environments? **(2) Robustness to representation methods:** Does BTD-sampling improve performance consistently across different latent encodings? **(3) Robustness of BTD to latent dimension:** Does BTD-sampling continue to perform well as the latent dimension grows? **(4) Combining task sampling strategies:** Can blending BTD-sampling with baseline task sampling lead to further improvements?

**Environments:** We conduct experiments on standard ZSRL benchmarks from ExoRL (Yarats et al., 2022), including Cheetah (run backward, walk backward, walk forward,

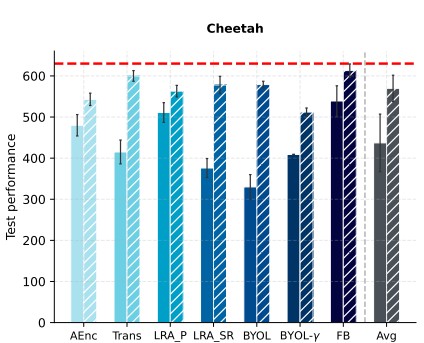
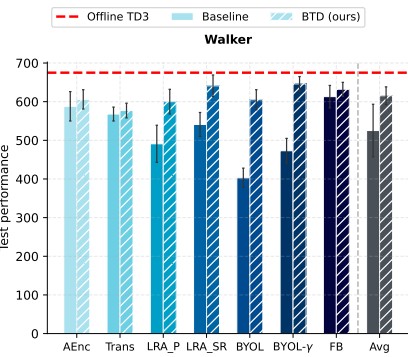
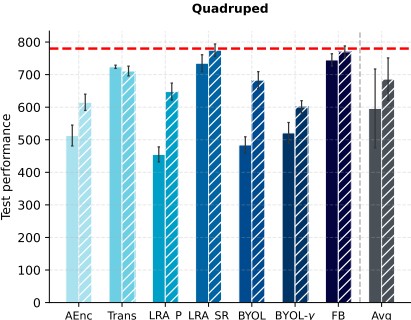

*Figure 1.* **Zero-shot performance comparison across task sampling strategies.** Results on Cheetah, Walker, and Quadruped for multiple representations show that BTD-sampling improves average performance and reduces sensitivity to representation choice. The horizontal red line indicates offline TD3 (oracle).

run forward), Walker (stand, walk, run, flip), and Quadruped (stand, walk, run, jump). For the main results, policies are trained using datasets collected by Random Network Distillation (Burda et al., 2018). Further details about the tasks are provided in Appendix C. Performance for each environment is computed as the average over its test tasks.

**Baselines:** We consider seven baselines:

• **Reconstruction and dynamics-based methods:** Autoencoder (**AEnc**) (Chen & Guo, 2023), Transition model (**Trans**), Lower-Rank Approximation of the transition probability (**LRA_P**) and of the successor measure (**LRA_SR**) (Touati et al., 2022), to learn state encoder $\phi(s)$, on top of which we train successor features.

• **Self-supervised representation learning methods:** **BYOL** (Grill et al., 2020) and **BYOL-$\gamma$** (Lawson et al., 2025) learn the state encoder $\phi(s)$ through latent-predictive learning, which we then use for learning successor features.

• **Forward-backward representations (FB)** (Touati & Ollivier, 2021).

For more details about the baselines, see Appendix D.

For each baseline, we compare two main variants that differ only in how reward functions are sampled during policy training: the original method using uniformly sampled task vectors (**Baseline**) and our proposed BTD-sampling variant (**BTD**).

**Implementation:** Across all experiments, we first train the state encoder $\phi(s)$, then we fix it and train the policy. Policy learning is performed using offline TD3 (Fujimoto et al., 2018). Network architectures and training hyperparameters are kept fixed across methods to isolate the effect of the task sampling strategy (see Appendix B.4). The latent task dimension is set to 50 by default, unless specified otherwise. The performance for one environment is computed as

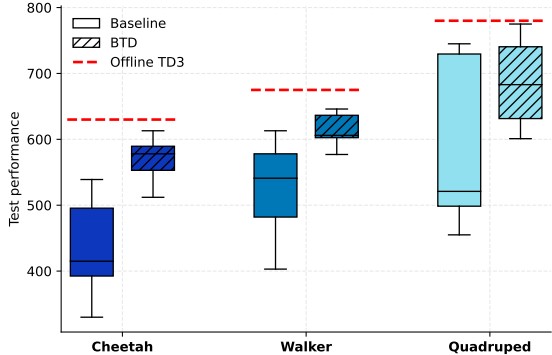

*Figure 2.* **Average test performance and variability of all baseline methods (solid) against their BTD-sampling counterparts (hatched) across three environments.** BTD-sampling consistently improves performance while significantly reducing variance, raising the lower bound of performance toward the offline TD3 oracle.

the average over its test tasks. For each test task, to estimate the corresponding task vector, we reveal the rewards of 5120 random states as in (Touati et al., 2022). All reported results are averaged over 5 seeds.

### 5.1. Overall Zero-Shot Performance

To assess the effectiveness of BTD-sampling for ZSRL, we compare baseline methods with their BTD-sampling counterparts across multiple environments and tasks.

**Results** As illustrated in Figure 1, our proposed BTD-sampling method consistently outperforms the corresponding baselines in nearly all experimental settings, achieving superior test performance in 13 of 15 tests and comparable performance in the remaining two, with an average improvement of **+20%**. This gain is particularly substantial in the Cheetah environment and when using **AEnc** and **LRA_P** to learn state embeddings. Moreover, BTD-sampling improves

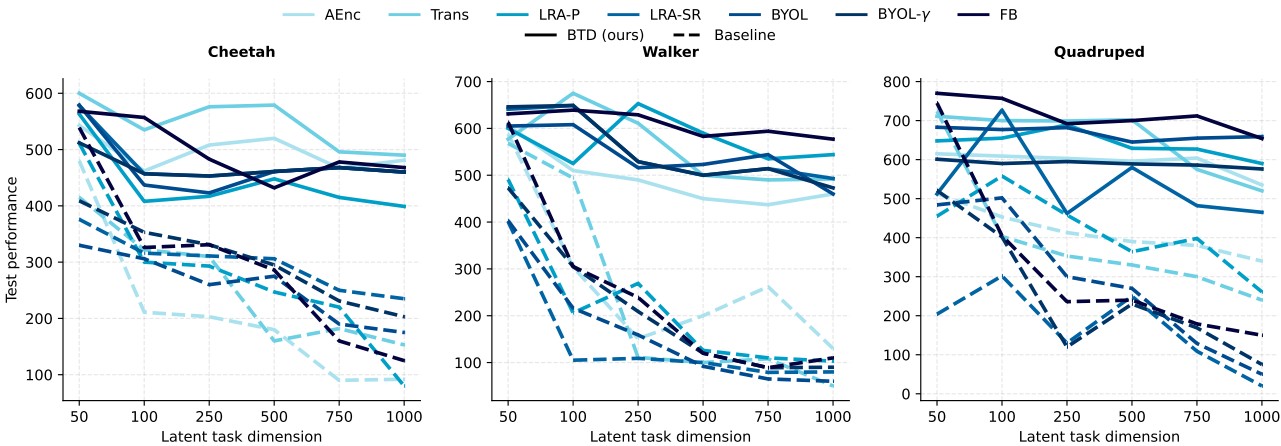

*Figure 3.* Test performance across a range of latent task dimensions $d$. Baseline performance drops in high dimensions as task vectors become increasingly orthogonal to the behavioral space, diluting the training signal. In contrast, BTD-sampling maintains robust performance by learning a task distribution that stays within the behavioral space, ensuring a consistent and informative training signal.

upon the state-of-the-art **FB** across all three environments. In general, these results show that BTD-sampling reliably improves performance in various locomotion tasks using various baseline methods.

## 5.2. Sensitivity to Representation Learning Methods

To evaluate the robustness of BTD-sampling across all baseline methods, we compute their average performance for each environment and compare the resulting distributions between the baseline and BTD-sampling, providing a measure of both overall performance and variability.

**Results** As illustrated in Figure 2, BTD-sampling consistently improves test performance while reducing the variance in all environments. The resulting test performance distributions are significantly more compact, indicating a higher degree of consistency compared to the baseline. Specifically, BTD-sampling significantly raises the lower bound of performance and improves all methods, including low-performing ones.

## 5.3. Effect of task space dimension

Proposition 4.1 predicts that baseline methods are increasingly likely to suffer from signal dilution as the latent task dimension grows, resulting in degraded performance. We evaluate this empirically and investigate whether BTD-sampling can maintain robust performance under the same conditions. To do so, we train policies using both baseline methods and BTD-sampling across a range of latent dimensions [50, 100, 250, 500, 750, 1000] and measure their performance in test tasks. This setup enables a direct comparison of the sensitivity of baselines to high-dimensional task spaces with the robustness offered by BTD-sampling.

**Results** Figure 3 shows that, consistent with Proposition 4.1, the baselines exhibit a sharp performance degradation as the dimension of the task space increases. In all three environments, the baselines struggle in high-dimensional settings, with scores often dropping rapidly as the dimension grows beyond 100. Particularly, in the Cheetah and Walker tasks, the performance of baselines collapses to near-failure performance at dimensions 750 and 1000.

This trend empirically confirms that uniform sampling approaches are highly sensitive to the dimensionality of the task space. As the dimension grows, uniformly sampled task vectors become increasingly likely to be orthogonal to the behavioral space, resulting in rewards that fail to discriminate between different behaviors.

In contrast, BTD-sampling demonstrates remarkable robustness to high-dimensional task spaces. While the performance of baselines decays rapidly, BTD-sampling maintains relatively stable and superior performance across the entire spectrum of latent dimensions. Even at dimensions as high as 1000, BTD-sampling models retain significantly higher scores compared to their baseline counterparts. These results indicate that BTD-sampling effectively mitigates the adverse effects of high-dimensional spaces. It does so by leveraging the offline dataset to align the task distribution with the behavioral space. This stabilizes the underlying baseline methods against the performance drops predicted by our theoretical analysis.

## 5.4. Combining task sampling strategies

We investigate whether combining BTD-sampling with existing task sampling strategies can further improve zero-shot performance. Specifically, we train policies using mixtures of tasks sampled from BTD and from a uniform distribution

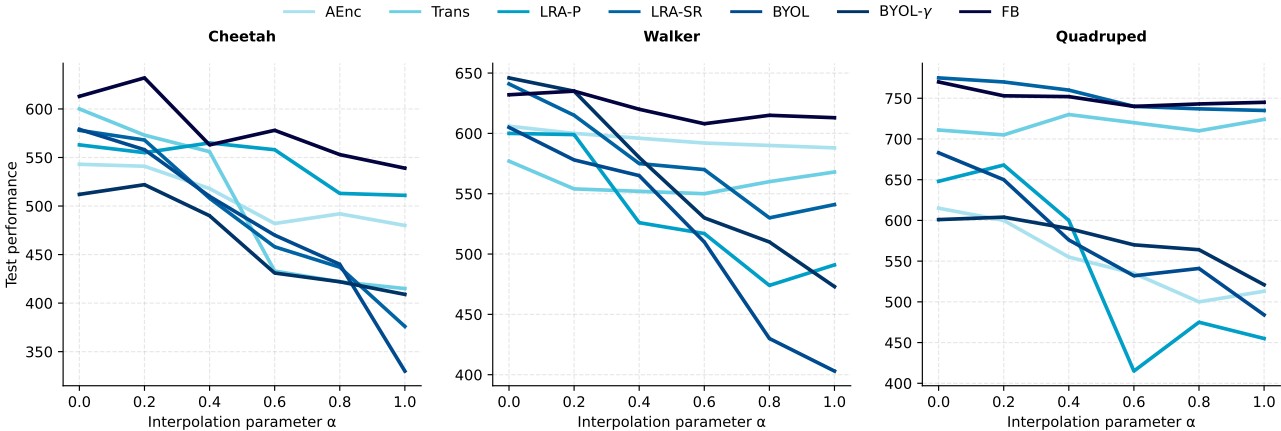

*Figure 4.* Test performance across Cheetah, Walker, and Quadruped as the interpolation parameter $\alpha$ varies from 0 (pure BTD-sampling) to 1 (pure uniform). Increasing the proportion of uniformly sampled tasks leads to performance degradation across almost all methods. This shows that uniform sampling dilutes the training signal rather than enhancing generalization. This confirms that aligning the task distribution with the behavioral space is essential for effective zero-shot transfer.

over the unit sphere $\mathrm{Unif}(\mathbb{S}^{d-1})$. By varying the mixture proportion, we aim to understand whether hybrid task sampling strategies can yield additional gains in generalization, or whether the introduction of uniformly random tasks interferes with the effectiveness of BTD-guided exploration.

**Results**   Figure 4 shows that performance consistently degrades as the interpolation parameter $\alpha$ increases. This trend is particularly pronounced in Cheetah and Walker, where performance drops across all tasks. In Quadruped, the effect is less evident, mainly because the baseline performance is already comparable to that of the BTD-sampling policy. Overall, these results indicate that incorporating uniformly random tasks does not improve robustness and instead harms policy performance. This underscores the importance of aligning the task distribution with the behavioral space to enable effective zero-shot generalization.

### 5.5. Ablation study

We validate key components in Appendix E: (1) comparisons of BTD-sampling to alternative sampling heuristics to determine whether the performance of BTD comes from the continuity of the distribution or from the dataset alone; (2) empirical verification of signal dilution, as demonstrated in Proposition 4.1; (3) effect of the number of GMM components on zero-shot performance; (4) effect of the number of subtrajectories $N_\tau$ on GMM estimation quality; and (5) visualization of the task space to assess how effective uniform sampling is at recovering behaviorally grounded tasks.

## 6. Conclusion

In this work, we identified a flaw in current offline zero-shot reinforcement learning (ZSRL): the reliance on uniform task sampling from the unit hypersphere during policy train-

ing. Our theoretical analysis reveals a "signal dilution" phenomenon in which reward variance vanishes as task dimensions increase, causing the learning signal to collapse. Crucially, we observed that the task distributions derived from the offline dataset is less affected by this growth, providing a robust signal regardless of the ambient dimension.

To back up this observation, we introduced the Behavioral Task Distribution (BTD) sampling, which extracts relevant task vectors from offline data to guide policy training. By grounding the task distribution in achievable behavioral spaces, BTD-sampling ensures that the learning process remains informative at any scale. Our empirical results demonstrate that BTD-sampling consistently outperforms standard baselines, achieving an average 20% improvement in zero-shot performance in the kind of physically relevant tasks one actually cares about in real-world deployment. Ultimately, our findings highlight that in offline ZSRL, defining which tasks to learn is as important as learning how to solve them, and that building on data-driven task distributions offers a scalable path toward more generalizable agents.

**Limitations.**   BTD is bounded by the behavioral support of the offline dataset: task vectors are extracted from observed trajectories, so feasible but rarely observed behaviors may be underrepresented in the learned distribution. Additionally, BTD inherits the linear reward assumption of SF/FB methods, as rewards outside the linear span of $\phi$ cannot be recovered regardless of the task sampling strategy. Finally, our experiments focus on locomotion tasks from the ExORL benchmark. Generalization to visually complex or contact-rich manipulation environments remains an open direction.

## Impact Statement

Our method operates entirely on pre-collected offline datasets and does not introduce new capabilities for harmful applications beyond those of existing offline RL methods. We do not anticipate significant negative societal impacts from this research.

## Acknowledgments

Experiments presented in this paper were carried out using the HPC resources of IDRIS under the allocation 2025-[AD011016374] made by GENCI. This work was supported by the Sorbonne Center for Artificial Intelligence (SCAI).

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

# A. Proof of Proposition 4.1

*Proof.* For a fixed task $z$, the variance of the returns is determined by the projection of the task vector onto the directions of variance in the behavioral space:

$$\text{Var}_\pi(J(\pi, z)) = \text{Var}_\pi\left((\psi^\pi)^\top z\right) \tag{18}$$

$$= \mathbb{E}_\pi\left[\left((\psi^\pi - \bar{\psi})^\top z\right)^2\right] \tag{19}$$

$$= \mathbb{E}_\pi\left[z^\top(\psi^\pi - \bar{\psi})(\psi^\pi - \bar{\psi})^\top z\right] \tag{20}$$

$$= z^\top\left(\mathbb{E}_\pi\left[(\psi^\pi - \bar{\psi})(\psi^\pi - \bar{\psi})^\top\right]\right)z \tag{21}$$

$$= z^\top\Sigma_\Psi z. \tag{22}$$

Now consider the expectation over the task distribution $z$. Using $\text{Tr}(ABC) = \text{Tr}(BCA)$ and linearity of expectation:

$$\mathbb{E}_z\left[z^\top\Sigma_\Psi z\right] = \mathbb{E}_z\left[\text{Tr}(z^\top\Sigma_\Psi z)\right] \tag{23}$$

$$= \mathbb{E}_z\left[\text{Tr}(\Sigma_\Psi zz^\top)\right] \tag{24}$$

$$= \text{Tr}\left(\Sigma_\Psi \mathbb{E}_z[zz^\top]\right). \tag{25}$$

For a uniform random vector on the unit hypersphere $\mathbb{S}^{d-1}$,

$$\mathbb{E}_z[zz^\top] = \frac{1}{d}I_d.$$

Substituting yields

$$\mathbb{E}_z[\text{Var}_\pi(J(\pi, z))] = \frac{1}{d}\text{Tr}(\Sigma_\Psi) = \frac{1}{d}\sum_{i=1}^{d}\lambda_i. \tag{26}$$

The goal is to prove that this converges to 0 as the dimension d grows.

The set of all possible behaviors (feature occupancies) is constrained by the environment's dynamics (e.g., gravity, collision limits, transition probabilities). We assume that the MDP $\mathcal{M}$ is fixed and independent of the representation dimension $d$: the complexity of the environment does not grow as $d$ increases. Consequently, the variance of the valid behaviors is concentrated in a limited number of effective dimensions. As we increase $d$, we add new dimensions that capture finer details of the behavioral space. However, because the total information content of the MDP is finite, the variance contributed by these additional dimensions must eventually vanish.

Formally, this implies that the sequence of eigenvalues $\{\lambda_i\}_{i=1}^{\infty}$ satisfy:

$$\lim_{i\to\infty}\lambda_i = 0. \tag{27}$$

This condition allows the total variance to grow, provided the incremental variance added by each new dimension diminishes. Using **Cauchy's Limit Theorem**:

$$\lim_{i\to\infty}\lambda_i = 0 \implies \lim_{d\to\infty}\frac{1}{d}\sum_{i=1}^{d}\lambda_i = 0. \tag{28}$$

This completes the proof.

$$\square$$

# B. Method details

## B.1. Architecture

We maintain consistent network architectures across all evaluated methods:

- The successor feature network $\phi(s)$ and backward representation $B(s)$ are implemented as feedforward networks with three 256-unit hidden layers that output $L_2$-normalized embeddings. We enforce the orthonormality of learned features through a regularization term.

- Our policy employs a TD3-based architecture (Fujimoto et al., 2018) with task-conditioned networks: the actor $\pi(s, z)$ uses parallel preprocessing streams (one for state features, another for task-conditioned states) with LayerNorm-Tanh initialization followed by ReLU activations, converging through a shared trunk network with Tanh output activation and truncated normal exploration ($\sigma = 0.2$). The critic uses twin Q-networks with analogous two-stream preprocessing of $(s, a, z)$ tuples for stable value estimation.

## B.2. Training the State Encoder

The state encoder $\phi(s)$ is trained using one of several representation learning objectives (see Appendix D).

The training process for FB representations is slightly different from the other methods. Although the FB objective simultaneously learns a backward embedding $B(s)$ and a policy $\pi$, we do not use them directly. Instead, to maintain a unified framework across all experimental baselines, we define the final state encoder as $\phi(s) = \mathbb{E}[BB^\top]^{-1}B(s)$ and train a new conditional policy from scratch using the reward $R_z(s) = \phi(s)^\top z$. We found that this approach does not hurt the performance of FB. Rather, it slightly improves it in the experiments, probably because the policy is trained on a fixed state representation, rather than a non-stationary one as in standard FB.

## B.3. Training the Policy

During policy training, instead of sampling 2048 tuples $(s_t, a_t, s_{t+1}, z)$, we first sample 32 task vectors and then sample 64 transitions $(s_t, a_t, s_{t+1})$ per task. This was more efficient in our experiments, as sampling multiple transitions per reward function allows the critic to more rapidly capture the structure of each reward landscape.

## B.4. Hyperparameters

*Table 1.* Hyperparameters for Dataset, Representation Learning, and Policy Training

| Category | Parameter | Value |
| --- | --- | --- |
| **Dataset** | Total Transitions | $10^7$ |
| | Number of Trajectories | $10^4$ |
| | Trajectory Length | $10^3$ |
| | Replay Buffer Size | $5 \times 10^6$ ($10 \times 10^6$ for maze) |
| **SF Training** | Gradient Updates | $10^5$ |
| | Batch Size | 1024 |
| | Learning Rate | $10^{-4}$ |
| | Target Update Rate ($\tau$) | 0.01 |
| | Discount Factor ($\gamma$) | 0.99 |
| | Orthonormality Weight | 1 |
| | Optimizer | Adam |
| **FB Training** | Gradient Updates | $2 \times 10^6$ |
| | Batch Size | 1024 |
| | Learning Rate | $10^{-4}$ |
| | Target Momentum | 0.99 |
| | Orthonormality Weight | 1 |
| | Optimizer | Adam |
| **Policy (TD3)** | Gradient Updates | $10^6$ |
| | Batch Size | 2048 |
| | Actor Learning Rate | $10^{-4}$ |
| | Critic Learning Rate | $10^{-4}$ |
| | Target Update Rate ($\tau$) | 0.01 |
| | Discount Factor ($\gamma$) | 0.99 |
| | Exploration Noise ($\sigma$) | 0.2 |
| | Policy Smoothing Truncation | 0.3 |
| | Optimizer | Adam |

## C. Environment details

This section details the physical dimensions and reward objectives for the continuous control environments used in our evaluations. All the environments considered in this paper are based on the DeepMind Control Suite (Tassa et al., 2018):

**Walker** Walker is a planar bipedal robot. The 24-dimensional state vector contains joint positions and velocities. Actions are 6-dimensional vectors. We evaluate performance across four distinct tasks at test time:

- **Stand:** Reward is a weighted combination of terms that encourage an upright torso orientation and maintain a minimum torso height.

- **Walk:** Reward includes a component that encourages a minimum forward velocity, capped at $4\,\mathrm{m/s}$.

- **Run:** Reward includes a component that encourages a higher minimum forward velocity, capped at $10\,\mathrm{m/s}$.

- **Flip:** Reward includes a component that encourages the agent to achieve a minimum angular momentum.

**Cheetah** The Cheetah is a planar biped designed. The 17-dimensional state vector contains positions and velocities. Actions are 6-dimensional vectors. We consider four velocity-based tasks at test time:

- **Walk:** Reward is linearly proportional to forward velocity, capped at $2\,\mathrm{m/s}$.

- **Run:** Reward is linearly proportional to forward velocity, capped at $10\,\mathrm{m/s}$.

- **Walk Backward:** Reward is linearly proportional to backward velocity, capped at $-2\,\mathrm{m/s}$.

- **Run Backward:** Reward is linearly proportional to backward velocity, capped at $-10\,\mathrm{m/s}$.

**Quadruped**   The Quadruped is a four-legged ant-like robot navigating in 3D space. The state and action spaces are 78-dimensional and 12-dimensional, respectively. We evaluate the following four tasks at test time:

- **Stand:** Reward encourages maintaining an upright torso posture.

- **Walk:** Reward includes a component that encourages a minimum forward torso velocity, capped at $0.5\,\mathrm{m/s}$.

- **Run:** Reward includes a component that encourages higher forward torso velocities, capped at $5\,\mathrm{m/s}$.

- **Jump:** Reward is proportional to the vertical displacement of the torso, it encourages the agent to maximize its peak height relative to the ground.

Across all environments, we utilize datasets from the ExORL (Yarats et al., 2022) benchmark collected via Random Network Distillation (RND) (Burda et al., 2018). Each dataset consists of $10^7$ total transitions, composed of $10^4$ trajectories with a length of $10^3$ steps each.

## D. Baselines details

This section provides objective functions for all baseline representation learning methods used in our experiments:

- **Autoencoder (AEnc)** (Chen & Guo, 2023) learns a compact representation $\phi(s)$ by minimizing the reconstruction error of the input state through an encoder-decoder architecture, capturing the most salient features of the state space. Here, $\phi$ is the encoder $\phi(s)$ and $f$ is the decoder.

$$\min_{f,\phi} \mathbb{E}_{s \sim \mathcal{D}} \left[ (f(\phi(s_t)) - s)^2 \right].$$

- **Transition Model (Trans)** learns representations by training a model to predict the next state $s_{t+1}$ given the current state-action pair $(s_t, a_t)$, thereby encoding the environment's local dynamics into the latent space. Here, $\phi$ is the encoder $\phi(s)$ and $f$ is the latent transition predictor.

$$\min_{f,\phi} \mathbb{E}_{(s_t,a_t,s_{t+1}) \sim \mathcal{D}} \left[ (f(\phi(s_t), a_t) - s_{t+1})^2 \right].$$

- **Lower-Rank Approximation of the transition probability (LRA_P)** (Touati et al., 2022) factorizes the one-step transition probability $P(s'|s,a)$ into a low-rank product $f(s,a)^\top \phi(s')$, effectively performing a spectral decomposition of the transition operator. Here, $f$ and $\phi$ represent the left and right singular vectors of the transition matrix.

$$\min_{f,\phi} \frac{1}{2} \mathbb{E}_{\substack{(s_t,a_t) \sim \mathcal{D} \\ s' \sim \mathcal{D}}} \left[ \left( (f(s_t, a_t)^\top \phi(s'))\right)^2 \right] - \mathbb{E}_{(s_t,a_t,s_{t+1}) \sim \mathcal{D}} \left[ f(s_t, a_t)^\top \phi(s_{t+1}) \right].$$

- **Lower-Rank Approximation of the successor measure (LRA_SR)** (Touati et al., 2022) learns a low-rank factorization of the successor measure $M(s, s')$ using temporal difference learning. Here, $f$ and $\phi$ are the learned factors that represent the state and its temporally extended features.

$$\min_{f,\phi} \mathbb{E}_{\substack{(s_t,s_{t+1}) \sim \mathcal{D} \\ s' \sim \mathcal{D}}} \left[ \left( f(s_t)^\top \phi(s') - \gamma f(s_{t+1})^\top \bar{\phi}(s') \right)^2 \right] - 2 \mathbb{E}_{(s_t,s_{t+1}) \sim \mathcal{D}} \left[ f(s_t)^\top \phi(s_{t+1}) \right].$$

- **Bootstrap Your Own Latent (BYOL)** (Grill et al., 2020) learns a latent space by predicting the next state representation from the current state-action pair using a predictor and a target network. Here, $\phi$ is the encoder and $\psi$ is the online predictor.

$$\min_{\phi,\psi} \mathbb{E}_{(s_t,a_t,s_{t+1}) \sim \mathcal{D}} \left[ \| \psi(\phi(s_t), a_t) - \text{sg}(\phi(s_{t+1})) \|_2^2 \right].$$

- **Bootstrap Your Own Latent with Bidirectional Prediction (BYOL-$\gamma$)** (Lawson et al., 2025) captures long-range temporal consistency by predicting future states at horizons sampled geometrically. Here, $\psi_f$ is the forward predictor towards future states and $\psi_b$ is the backward predictor towards past states.

$$\min_{\phi,\psi_f,\psi_b} \mathbb{E}_{\substack{(s_t,a) \sim \mathcal{D} \\ k \sim \text{Geom}(1-\gamma)}} \left[ \| \psi_f(\phi(s_t), a) - \text{sg}(\phi(s_{t+k})) \|_2^2 + \| \psi_b(\text{sg}(\phi(s_{t+k}))) - \phi(s_t) \|_2^2 \right].$$

- **Forward-Backward Representations (FB)** (Touati & Ollivier, 2021) jointly learns forward $F(s, a, z)$ and backward $B(s)$ representations to estimate successor measures for any arbitrary policy $\pi_z$. Here, $F$ predicts future behavior under a goal/task $z$, while $B$ encodes state-specific features.

$$\min_{F,B} \mathbb{E}_{\substack{(s_t,a_t,s_{t+1}) \sim \mathcal{D} \\ s' \sim \mathcal{D}}} \left[ \left( F(s_t, a_t, z)^\top B(s') - \gamma F(s_{t+1}, \pi_z(s_{t+1}), z)^\top \bar{B}(s') \right)^2 \right] - 2 \mathbb{E}_{(s_t,a_t,s_{t+1}) \sim \mathcal{D}} \left[ F(s_t, a_t, z)^\top B(s_{t+1}) \right].$$

## E. Additional Experiments: Ablation Study

This section presents a series of ablation studies designed to isolate and validate the key design choices underlying BTD-sampling. We investigate (1) whether performance gains arise from continuous density modeling rather than from heuristic task extraction, (2) empirically validate the signal-dilution phenomenon predicted in Proposition 4.1, (3) analyze the sensitivity to the number of mixture components, and (4) visualize the structure of the learned task space.

*Table 2.* **Ablation of task sampling strategies.** Zero-shot test performance comparison of BTD against heuristic baselines. BTD-sampling outperforms (Subtraj) sampling by interpolating the task space, while (full traj) sampling fails, as full trajectories are dominated by failures where the useful signal is minority.

| Sampling Strategy | Cheetah | Walker | Quadruped |
|---|---|---|---|
| Uniform | $464 \pm 60$ | $560 \pm 42$ | $634 \pm 124$ |
| Full trajectory | $482 \pm 37$ | $488 \pm 11$ | $421 \pm 30$ |
| Subtrajectory | $512 \pm 24$ | $579 \pm 25$ | $678 \pm 43$ |
| **BTD-sampling (ours)** | $\mathbf{579 \pm 25}$ | $\mathbf{611 \pm 23}$ | $\mathbf{704 \pm 64}$ |

### E.1. Comparison to Heuristic Task Sampling

This experiment investigates whether the gains observed with BTD-sampling stem from its continuous density modeling or merely from extracting tasks from the offline dataset through $p_{data}$. We compare BTD-sampling with two heuristic baselines: (i) **Subtrajectory sampling**, where tasks are directly sampled from $p_{data}$; (ii) **Full trajectory sampling**, where task vectors are computed from entire trajectories.

**Results:** BTD-sampling consistently outperforms **Subtrajectory sampling** across all benchmarks (Table 2). Although both rely on $p_{data}$, BTD enables interpolation between tasks and yields more diverse tasks.

In contrast, **Full trajectory sampling** performs poorly, particularly on Quadruped. As full trajectories contain multiple behaviors over time, they can produce task vectors where meaningful primitive behaviors are drowned into more noisy signal. In environments where recovery from failure is difficult, such as Quadruped, the resulting task vectors are dominated by failure and provide poor training reward functions.

### E.2. Empirical Validation of Signal Dilution

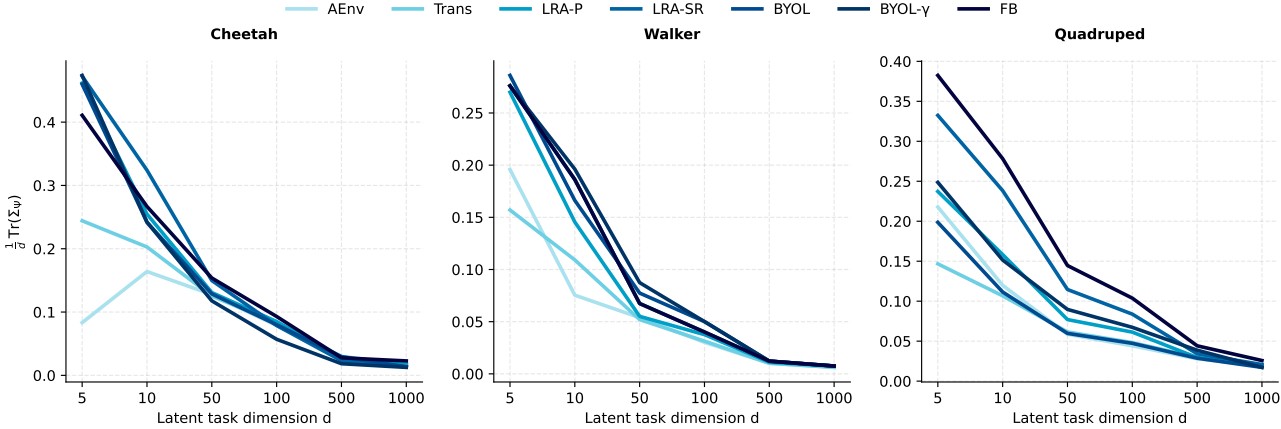

*Figure 5.* **Signal dilution under increasing feature dimension.** The normalized trace of the empirical feature-occupancy covariance $(1/d)\mathrm{Tr}(\Sigma_\Psi)$ decays rapidly as the feature dimension $d$ increases across environments and baselines.

Proposition 4.1 characterizes the decay of the expected return variance when the tasks are uniformly sampled. The theoretical quantity $\Psi$ represents the set of all possible feature occupancies under the MDP, which is not directly observable. We therefore construct an empirical counterpart $\tilde{\Psi}$ from the offline dataset by sampling $100,000$ random subtrajectories $\tau$ and computing their discounted feature occupancies:

$$\tilde{\Psi} = \{\tilde{\psi}^\tau \mid \tau \in \mathcal{D}_{sub}\}, \qquad \tilde{\psi}^\tau = \sum_{t=0}^{|\tau|} \gamma^t \phi(s_t). \tag{29}$$

For each environment, each baseline method, and each feature dimension $d \in \{5, 10, 50, 100, 500, 1000\}$, we train the corresponding state representation $\phi(s)$. We then compute $\tilde{\Psi}$ and its covariance $\Sigma_{\tilde{\Psi}}$, and report the upper bound quantity $(1/d)\mathrm{Tr}(\Sigma_{\tilde{\Psi}})$.

**Results:** Figure 5 shows a rapid exponential decay of $(1/d)\mathrm{Tr}(\Sigma_{\tilde{\Psi}})$ as $d$ increases, confirming the signal-dilution effect predicted by Proposition 4.1. This confirms that the vanishing variance is not an artifact of a specific representation or environment, but a fundamental consequence of uniform sampling.

### E.3. Effect of number of components in the GMM

This ablation study examines the impact of the number of Gaussian Mixture Model (GMM) components on the test performance across the benchmark environments and all the baselines, across a range of 5 to 200 GMM components.

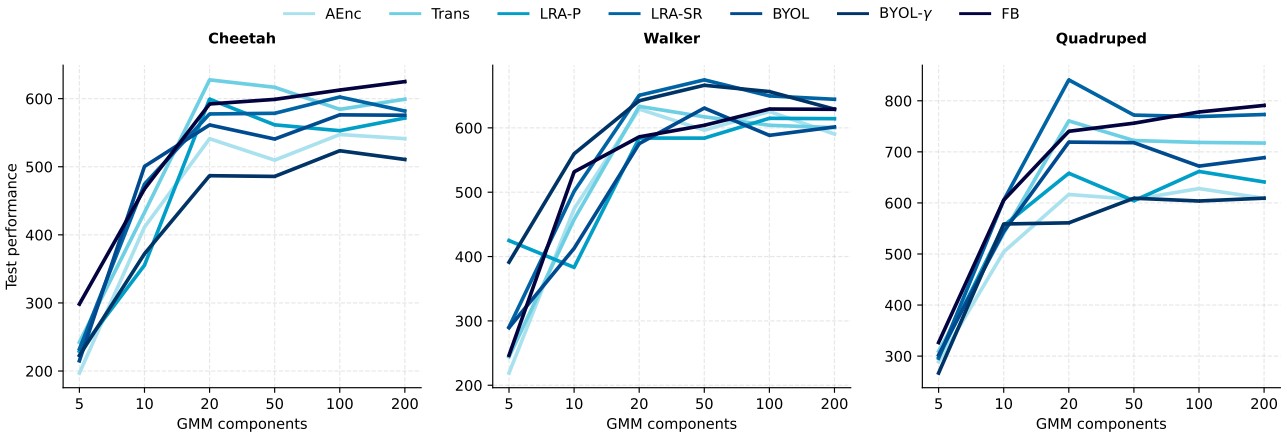

*Figure 6.* **Effect of the number of GMM components on test performance.** Performance improves rapidly with more components up to about 20, after which gains generally plateau across environments and baselines.

**Results:** Across all environments and baseline methods, we observe a consistent trend: performance increases rapidly until approximately 20 components, after which gains diminish. This indicates that the behavioral task distribution can be effectively captured with a relatively small number of components (20), and the exact choice beyond this point is not critical.

### E.4. Effect of the Number of Subtrajectories $N_\tau$

This ablation study examines the impact of the number of subtrajectories $N_\tau$ sampled from the dataset to fit the GMM, across a range from 10 to $10^6$. We use FB (BTD) as the baseline method.

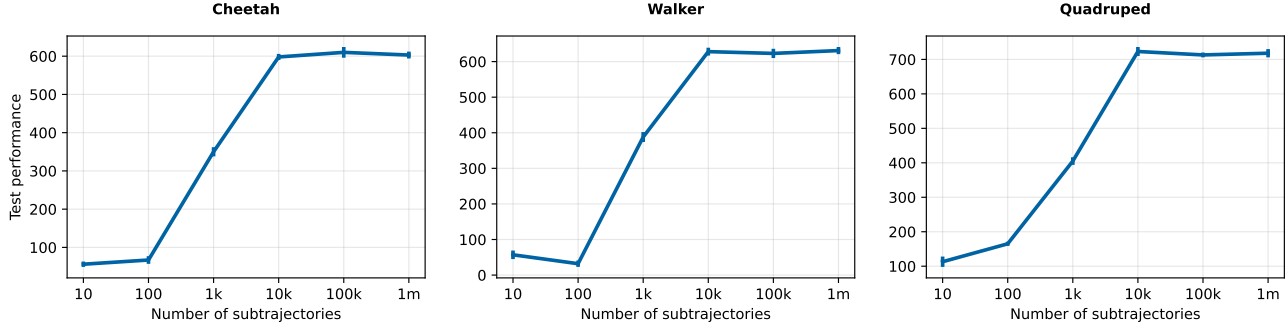

*Figure 7.* **Effect of the number of subtrajectories $N_\tau$ on test performance.** Performance improves rapidly with more subtrajectories up to about $10k$, after which gains plateau across environments, indicating sufficient coverage of the behavioral task distribution.

**Results:** Across all three environments, performance improves consistently as $N_\tau$ increases, then saturates around $N_\tau = 10,000$, beyond which additional subtrajectories yield negligible gains. Small values of $N_\tau$ produce poor estimates of the behavioral task distribution and consequently low performance, while $N_\tau = 10,000$ provides sufficient coverage of the behavioral space. In practice, we use $N_\tau = 10,000$ as the default, as it offers a good trade-off between estimation quality and computational cost.

### E.5. Visualization of the Task Space

We use UMAP projections to visualize the alignment between the empirical task distribution ($z \sim p_{data}$), extracted directly from the offline dataset, and the standard uniform prior ($z \sim \text{Unif}(S^{d-1})$). By comparing these distributions, we evaluate whether the uniform task sampling effectively captures the behavioral task distribution.

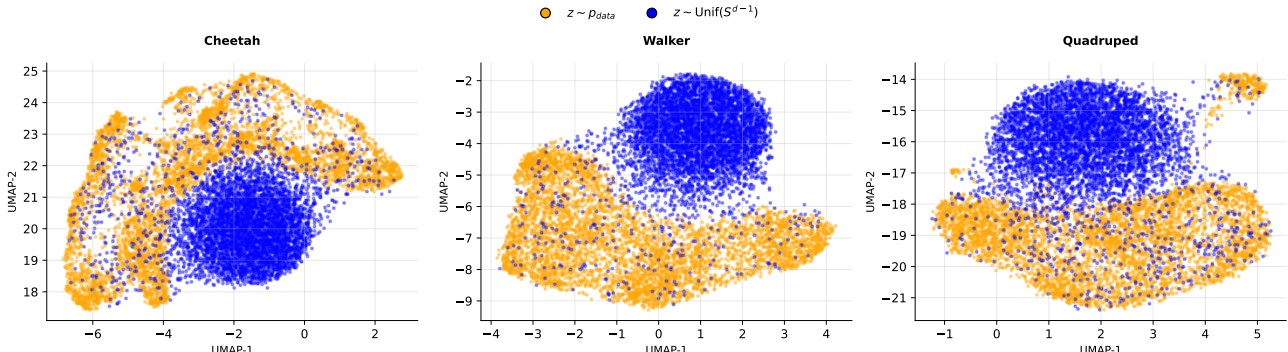

*Figure 8.* **UMAP visualization of task distributions.** Comparison between empirical task vectors $z \sim p_{data}$ and uniformly sampled task vectors $z \sim \text{Unif}(\mathbb{S}^{(}d-1)$, showing limited overlap.

**Results:** The visualizations in Figure 8 reveal a distributional shift between the uniform prior and the data-driven task vectors. While the uniform prior theoretically spans the entire hypersphere, our results show only marginal overlap with the empirical task manifold. This shows that the behavioral task vectors reside in a small region of the hypersphere that cannot be efficiently recovered using uniform sampling.

