# OpenReview forum: "Improving Zero-Shot Offline RL via Behavioral Task Sampling"
_ICML.cc/2026/Conference — ICML 2026 regular_

### Official Review · Reviewer_fEAH · 2026-02-19

**Soundness:** 4
**Presentation:** 4
**Significance:** 3
**Originality:** 3
**Overall Recommendation:** 5
**Confidence:** 3

**Summary:**

The contribution of this paper, to my understanding, is a new task sampling method for zero shot offline reinforcement learning, which builds a distribution over tasks from the dataset and samples from that distribution. In addition, this paper demonstrates a theoretical justification why uniform task sampling will struggle, especially as the latent dimension increases. This paper proposes that sampling tasks informed by the data distribution would improve the quality of learning, and they demonstrate this empirically by applying their sampling across a variety of baselines on some benchmark tasks.

**Compliance With Llm Reviewing Policy:**

Affirmed.

**Final Justification:**

I am happy to recommend acceptance for this paper.

**Key Questions For Authors:**

1. Could you please explain precisely what is your contribution, and what is novel about your method? The content is within the paper but I believe it could benefit from a more precise statement.
2. Are there any related works which are doing non-uniform sampling of the task vectors?

**Limitations:**

Yes

**Strengths And Weaknesses:**

The main strength of this paper is the innovative idea and thorough evaluation. The critical gap and problem are well explained, followed by a theoretically justified motivation for their method. The contribution is clear, and the evaluation aligns very well with that contribution. I commend the authors for answering several key questions that a reader would have with their evaluation. I particularly liked the analysis over the latent dimension and interpolation of sampling methods. I found the overall argument of the paper to be convincing. Thank you very much for your submission, which I enjoyed reading.

The main weakness of this paper is that there is no explicit statement of contribution, this can improve the paper by making very clear that which is novel and which you contribute. The related work could also do better at answering the question: what is novel about your method? The text makes it clear what you are doing, but I believe it is missing further explanation of what has not been done. What other methods do non-uniform task sampling? What are their approaches? If no other method has applied non-uniform task sampling, then this can be explained better in the literature review. If other methods have done this in the zero-shot reinforcement learning domain, then it is very important to explain why your method is different and novel. For example, could you please comment on whether the following paper would merit inclusion in the related works, based on how they do task sampling from a non-uniform distribution:
	* Zero-Shot Adaptation of Behavioral Foundation Models to Unseen Dynamics by Bobrin et al., (https://arxiv.org/pdf/2505.13150) please refer to Section 3.3 where sampling of task vectors follows a von Mises-Fisher distribution.

---

> ### Author Rebuttal · Authors · 2026-03-30
>
> We sincerely thank the reviewer for their thoughtful and encouraging feedback. We are thrilled that they found our paper enjoyable to read, and that our idea, thorough evaluation, and analyses were convincing. We also appreciate the suggestions on clarifying our contributions and positioning relative to prior work, which we address below.
>
>
> # Concerning the suggested paper:
> We thank the reviewer for pointing out the relevant paper by Bobrin et al.. We agree that it should be included in the related work section, in Section 2.1. We will definitely add it in the revised version.
>
> # Q1. Our contribution:
> In short, our contribution is to:
> - Identify uniform task sampling as a fundamental limitation in offline zero-shot RL
> - Theoretically characterize the failure mode of uniform task sampling
> - Introduce a simple, plug-and-play behavioral task sampling scheme
> - Demonstrate an average improvement of 20% in zero-shot performance compared to the baselines across methods, environments, and dimensions.
>
>
> We appreciate the reviewer’s suggestion and will make our contributions and novelty statement more explicit in the introduction of the revised version.
>
> # Q2. Other tasks using non-uniform sampling:
> To the best of our knowledge, most prior works rely on uniform sampling of task vectors. An exception is FB-CPR [3], which augments uniformly sampled task vectors with additional ones constructed from expert trajectories, where each task vector is defined as the average state representation over a trajectory. While this approach is primarily designed for distilling expert datasets into a policy, it is nevertheless relevant to our work. We will include and position this work in the related work section in the revised version.
>
> [3] Tirinzoni, A., Touati, A., Farebrother, J., Guzek, M., Kanervisto, A., Xu, Y., Lazaric, A. and Pirotta, M., 2025. Zero-shot whole-body humanoid control via behavioral foundation models. arXiv preprint arXiv:2504.11054.
>
> # Conclusion:
> We once again thank the reviewer for their constructive suggestions. Their feedback has provided valuable guidance that will help us strengthen the clarity and impact of the final version of the paper.

---

> > ### Author Rebuttal · Reviewer_fEAH · 2026-04-03
> >
> > I thank the authors for their responses, they have addressed my comments throughly.

---

### Official Review · Reviewer_5s42 · 2026-02-23

**Soundness:** 3
**Presentation:** 2
**Significance:** 2
**Originality:** 3
**Overall Recommendation:** 4
**Confidence:** 2

**Summary:**

This paper addresses offline zero-shot RL (ZSRL), where an agent must optimize unseen reward functions without further environment interaction. The manuscript argues that a key bottleneck in existing ZSRL pipelines (e.g., Successor Features and Forward–Backward representations) is not only the quality of the learned representation $\phi(s)$, but also how task vectors $z$ are sampled during policy training. This paper replaces uniform task sampling with a dataset-driven task distribution, constructing task vectors directly from the offline dataset by extracting sub-trajectories $\tau$, computing an empirical feature occupancy $\tilde\psi_\tau=\sum_t \gamma^t \phi(s_t)$, and normalizing to obtain $z_\tau=\tilde\psi_\tau/\|\tilde\psi_\tau\|$. This yields an empirical task set $p_{\text{data}}=\{z_\tau\}$, from which the authors fit a parametric Behavioral Task Distribution (BTD)—implemented as a GMM—and then sample $z\sim p_\theta(z) $ during policy training. The method is evaluated on ExoRL continuous-control benchmarks (Cheetah/Walker/Quadruped with multiple test tasks each), with policies trained via offline TD3 and multiple representation learning backbones (AEnc, Trans, LRA-P/LRA-SR, BYOL, BYOL-\gamma, FB). Across baselines, BTD sampling yields an average ~+20% improvement in test performance and reduces variance.

**Compliance With Llm Reviewing Policy:**

Affirmed.

**Final Justification:**

My Concerns are addressed.

**Key Questions For Authors:**

1. Since BTD is extracted from behaviors present in the offline dataset, how does the method behave when test rewards incentivize behaviors that are rare (but feasible) under the dataset policy? Is there a regime where BTD overfits to dataset behavior and hurts zero-shot performance?
2. What is the sensitivity to the sub-trajectory length distribution used to build $p_{\text{data}}$? Intuitively, short segments may emphasize local dynamics while long segments may emphasize long-horizon behaviors; does one dominate in practice?
3. Why a GMM rather thana hyperspherical von Mises–Fisher mixture, kernel density estimate, or normalizing flow on the sphere? Is the GMM simply acting as a smoothing mechanism over $p_{\text{data}}$, and how sensitive is performance to the number of mixture components?

**Limitations:**

The paper does include a brief limitation discussion. One limitation could be: BTD is bounded by dataset behavior support. Because tasks are extracted from observed sub-trajectories, the learned task distribution may underrepresent feasible but rarely observed behaviors. This could limit performance on test tasks that require such behaviors, even if they remain in the linear span of $\phi$.

**Strengths And Weaknesses:**

Strengths:
1. The proposal is minimalistic: do not change the algorithm; change the task sampling distribution. The method is compatible with both SF and FB-style pipelines and does not require modifying the representation learning objective, which makes the idea easy to adopt broadly.
2. Proposition 4.1 formalizes why uniform sampling becomes problematic as task dimension grows (vanishing variance / weak discrimination across policies), and the experiments explicitly verify dimension sensitivity and show BTD robustness in high-dimensional task spaces.

Weaknesses:
1. While the method claims algorithm-agnosticism, BTD is defined through $\phi(s)$ and sub-trajectory occupancies. If $\phi$ poorly captures semantics, or if the dataset behavior policies fail to cover relevant modes, the learned task distribution could bias training toward narrow behavior classes and harm generalization to certain test rewards. This concern is not fully stress-tested in the main text.
2. The Appendix proof for vanishing variance relies on a qualitative claim that the MDP has finite information content so eigenvalues $\lambda_i\to 0$. While this may hold in practical settings, it is not a standard assumption stated in the main body, and it is unclear what counterexamples might look like (e.g., expanding feature maps with non-vanishing variance directions).

---

> ### Author Rebuttal · Authors · 2026-03-31
>
> We thank the reviewer for their thoughtful feedback and positive assessment of our work. Below, we address their concerns.
> # W1. Poor $\psi$ or limited data coverage:
> BTD aims to cover test rewards achievable under the environment dynamics and representable in the linear span of $\psi$, as in SF/FB methods. Rewards outside this span cannot be solved by either baseline or BTD, so the limitation is in the representation, not the task sampling strategy.  Regarding dataset coverage, if relevant modes are missing, BTD will reflect this and focus training on regions with meaningful policy gradients in contrast to uniform sampling. Note that zero-shot RL literature always assumes sufficient dataset coverage.
> # W2. The assumption of finite information content:
> It's true the claim is stated qualitatively. We will address this by explicitly adding this assumption to the main body: The MDP is fixed and independent of the representation dimension. This assumption is natural in RL, as environments are independent of such external factors.
> Regarding expanding feature maps with non-vanishing variance, there are two cases: either (1) these directions encode truly new behaviors, which requires MDP complexity to grow with $d$ which is excluded by our assumption, or (2) variance spreads across dimensions without changing the MDP, keeping total variance $\operatorname{Tr}(\Sigma_\Psi)$ bounded by $C$, so $\lim_{d \to \infty} \frac{1}{d}\operatorname{Tr}(\Sigma_\Psi) \le \lim_{d \to \infty} \frac{C}{d} = 0$ remains. Thus the proof’s core mechanism holds.
>
> We will update the final part of the proof and expand the limitations section to acknowledge the potential existence of counterexamples outside our scope.
> # Q1. Rare test tasks and overfitting:
> BTD extracts task vectors from sub-trajectories of varying lengths, providing implicit coverage of rare behaviors. Even if complete "jump" or "flip" trajectories are rare under the dataset policy, $p_{data}$ still reflects short sub-trajectories capturing the corresponding behavior primitives. The GMM interpolates over these vectors, assigning non-zero probability mass to rare directions. Also, since uniform sampling assigns equal probability to all directions, it does not offer an advantage over BTD for rare test tasks either. Regarding overfitting: BTD's concentration on a subset of task directions is not overfitting but the correct inductive bias, since test tasks are physically grounded, the relevant task space is the behavioral space $\Psi$ that BTD targets.
> We keep testing BTD in cases where test tasks are decoupled from offline transitions for future work.
> # Q2. Sensitivity to the Sub-Trajectory
> In practice, we uniformly vary subtrajectory lengths from 1 to full length (1000), so no length dominates. Using a random mix of lengths works best (Appendix E.1).
> ## Zero-shot performance using subtrajectories of varying lengths:
> We use **Low-Rank Adaptation of Transition Probability (LRA_P)** to learn state representations and train a policy on top of it. Other baselines will be included in the final revision.
> Subtrajectory Length|5|10|50|100|500|1000|Unif(1,1000)
> -|-|-|-|-|-|-|-
> Cheetah|528±14|520±17|552±12|550±19|512±15|499±11|**569±16**
> Walker|548±18|569±22|580±17|572±24|533±19|501±16|**611±20**
> Quadruped|615±25|622±21|639±28|611±15|466±29|449±22|**704±30**
>
> **Analysis:**
> Performance improves with lengths in 50–100, very short or long lengths underperform due to limited context or excessive noise. Mixed lengths consistently yield the best zero-shot results, showing the benefit of diverse temporal information.
> # Q3. Other density estimators:
> We chose GMM for its simplicity and lightweight training. Results comparing GMM, vMF, and KDE are below. Normalizing flows were excluded due to short rebuttal period and high training cost which would compromise BTD’s plug-in nature. LRA_P is used as method.
> ||Cheetah|Walker|Quadruped
> -|-|-|-
> GMM|579±25|611±23|704±64
> vMF|533±50|552±42|678±33
> KDE|528±9|586±11|671±13
>
> Analysis: GMM outperforms vMF despite vMF being natural for spherical data, because its covariance matrices capture correlations between task dimensions that vMF’s scalar concentration parameter cannot. It also surpasses KDE, whose non-parametric structure limits expressivity. Task vectors are normalized to unit norm when sampling.
> ## Smoothing and sensitivity to components:
> GMM acts as a smoothing mechanism, which is crucial for improving zero-shot performance by generalizing beyond observed tasks, increasing diversity. Note that sampling from GMM consistently outperforms directly sampling $p_{data}$ (Appendix E.1).
> We provide an ablation on the number of GMM components for zero-shot performance in Appendix E.3. Performance rises quickly up to 20 components, then plateaus, indicating 20 components are enough to effectively capture the behavioral task distribution.
>
> We appreciate the reviewer’s insightful comments and limitations, and will improve the final version of the paper.

---

> > ### Author Rebuttal · Reviewer_5s42 · 2026-04-03
> >
> > Thank the authors for their response. My Concerns are addressed.

---

### Official Review · Reviewer_ysQX · 2026-03-04

**Soundness:** 3
**Presentation:** 3
**Significance:** 4
**Originality:** 4
**Overall Recommendation:** 5
**Confidence:** 2

**Summary:**

This work proposes an improvement to the standard approach to offline zero-shot reinforcement learning. Standard approaches estimate reward functions using learned state embeddings and randomly sampled task vectors. The authors demonstrate issues with randomly sampled task vectors and instead propose generating task vectors from the offline dataset. They provide theoretical backing for their work, as well as empirical results showing their method’s improvement over the baseline in average performance, lower performance variance, and better scalability to higher dimensional task vectors.

**Compliance With Llm Reviewing Policy:**

Affirmed.

**Final Justification:**

The rebuttal addressed my concerns and I have raised my score as indicated in response.

**Key Questions For Authors:**

1. How well does this method scale to more complex RL environments?
2. If standard offline RL outperforms these methods, and you need some form of offline dataset to produce the task vectors, in what situations would you use offline zero-shot RL instead of standard offline RL? What benefits does it provide in exchange for the drop in performance vs standard offline RL?
3. Are there any issues with z relying on phi, given that the reward estimation is $\phi(s)^Tz?$

**Limitations:**

yes

**Strengths And Weaknesses:**

Strengths:
1. The proposed improvement is simple, logical, and is strongly backed by theoretical and empirical results.
2. The results are explained well and the graphs are easy to follow.

Weaknesses:
1. Practical applications and use-cases are unclear. See Question 2 for specific questions on this point.
2. While the main idea proposed is quite simple and easy to understand, the paper feels very dense and difficult to follow. Though I admit it may be due to my own lack of familiarity with the background works.
3. (minor issue) The colors in Figure 3 are difficult to distinguish, especially the darker shades.

---

> ### Author Rebuttal · Authors · 2026-03-30
>
> We greatly appreciate the reviewer’s careful reading and insightful comments. It is encouraging to see that the simplicity and soundness of our proposed improvement were recognized, along with the clarity of our figures and results. Such acknowledgment reinforces the accessibility and practical significance of our approach.
> Below, we address questions regarding scaling to more complex environments, practical advantages of zero-shot RL, and the reliance of task vectors on $\phi(s)$.
>
> # Q1. More complex environments:
> In the paper we already tested our method on 3 DMC tasks including **Quadupred** having 78-dim state and 12-dim action, (often considered complex in the zero-shot RL literature). Upon the reviewer’s request, we also test our method on 3 new environments: Walker RGB and Cheetah RGB from Deepmind Control Suite, where the state is represented by a 64x64x3 image of the agent, as well as Cube-single from Ogbench, representing a manipulation environment.
>
> For the visual tasks, we use a simple CNN encoder with 4 convolutional layers. To incorporate temporal information, each state consists of a stack of 3 consecutive frames. We used the same settings as in the state-vector-based environments. Additional hyperparameter tuning can improve results.
>
> The results are in the table below:
> |  | Cheetah-RGB | Walker-RGB | Cube-single |
> |---|---|---|---|
> | Aenc | 323 ± 13 | 258 ± 18 | 22 ± 6 |
> | Aenc (BTD) | 512 ± 8 | 382 ± 5 | 36 ± 3 |
> | Trans | 456 ± 18 | 318 ± 18 | 15 ± 7 |
> | Trans (BTD) | 512 ± 9 | 383 ± 6 | 33 ± 3 |
> | LRA_P | 493 ± 13 | 315 ± 16 | 34 ± 5 |
> | LRA_P (BTD) | 541 ± 7 | 453 ± 8 | 37 ± 2 |
> | LRA_SR | 503 ± 16 | 345 ± 17 | 37 ± 5 |
> | LRA_SR (BTD) | 523 ± 6 | 412 ± 10 | 39 ± 2 |
> | BYOL | 472 ± 12 | 532 ± 15 | 38 ± 4 |
> | BYOL (BTD) | 576 ± 8 | 571 ± 8 | 42 ± 2 |
> | BYOL-$\gamma$ | 465 ± 15 | 552 ± 16 | 40 ± 4 |
> | BYOL-$\gamma$  (BTD) | 562 ± 9 | 582 ± 7 | 43 ± 2 |
> | FB | 621 ± 24 | 454 ± 6 | 49 ± 3 |
> | FB (BTD) | 643 ± 9 | 522 ± 10 | 53 ± 1 |
> | **Average** | **475 ± 81** | **396 ± 107** | **34 ± 11** |
> | **Average (BTD)** | **552 ± 43** | **472 ± 80** | **41 ± 6** |
>
>
> **Analysis:**
> The results consistently support the paper’s main claim that replacing uniform sampling with BTD delivers constant gains across all representations and environments, improving average performance while reducing variance, indicating that BTD stabilizes training by providing a more informative signal. This reinforces the original conclusions of the paper.
>
> # Q2. Standard RL vs Zero-shot RL:
>
> Standard offline RL trains a separate policy for each task, requiring full retraining whenever a new task is introduced. Zero-shot RL trains once and generalizes to any new task at inference time with no additional training or environment interaction. This distinction matters in practical scenarios, for example, when the task is specified at deployment time and cannot be anticipated during training, or when the number of tasks is large, making per-task retraining computationally prohibitive.
> Zero-shot RL trades some performance for flexibility and instant adaptation capabilities.
>
> # Q3. Reliance of the task vector $z$ on $\phi$:
> We thank the reviewer for this question. Extracting $z$ from $\phi$ is not problematic because the same $\phi$ is used at test time to estimate the task vector: in SF, $z_{\text{test}} = \arg\min_z \mathbb{E}[(R_{\text{test}}(s) - \phi(s)^\top z)^2]$, and in FB, $z_{\text{test}} = \mathbb{E}[R_{\text{test}}(s) \cdot B(s)]$ with $\phi(s) = \mathbb{E}[BB^\top]^{-1} B(s) $. In both cases, $z_{\text{test}}$ is recovered through $\phi$, meaning training and evaluation are fully symmetric with respect to $\phi$. There is no additional assumption or risk introduced by BTD, as it simply operates within the same features defined by $\psi$ that the method already relies on at test time. If $\phi$ is rich, BTD extracts rich tasks. If $\phi$ is limited, both BTD and the baseline are equally limited. The dependence on $\phi$ is therefore not a flaw specific to our method, but a shared property of the entire SF/FB family.
>
>
> # W3. Colors low contrast:
> We will update the figures' color palette to increase contrast and improve clarity in the final version.
>
> # Conclusion:
> We appreciate the reviewer’s insightful suggestions once again. These comments and the clarifications they prompted will help us further improve the final version of the paper.

---

> > ### Author Rebuttal · Reviewer_ysQX · 2026-03-31
> >
> > This resolves my issues. I have increased the significance score and the overall recommendation accordingly.

---

### Official Review · Reviewer_xNGi · 2026-03-13

**Soundness:** 3
**Presentation:** 3
**Significance:** 3
**Originality:** 3
**Overall Recommendation:** 4
**Confidence:** 3

**Summary:**

This paper studies offline zero-shot reinforcement learning problems and focuses on the task (latent) sampling strategy used to train the task-conditioned policies. The main claim is that existing methods, such as successor features and forward-backward representations, typically sample task vectors uniformly from the unit sphere. This might produce weak training signals in high-dimensional latent spaces because most sampled tasks are poorly aligned with behaviors that are actually achievable in the environment. To address this, the paper proposes Behavioral Task Distribution (BTD) sampling, which extracts task vectors from discounted feature occupancies of subtrajectories in the offline dataset. Empirically, the paper reports consistent gains across several ExORL locomotion benchmarks and several representation-learning baselines, with around 20% average improvement over the uniform task sampling strategy.

**Compliance With Llm Reviewing Policy:**

Affirmed.

**Final Justification:**

The rebuttal addressed my main concerns. Meanwhile, I encourage the author to incorporate new experimental results and clarifications.

The current rating can positively support the work, and I will update the presentation score accordingly.

**Key Questions For Authors:**

- In the introduction, the paper motivates the issue of sampling from the uniform VMF distribution as “In high-dimensional settings, due to the concentration of measure phenomenon (Ledoux, 2001), uniformly sampled tasks tend to be almost orthogonal to the behaviors the agent can produce.” It is not clear what this concentration of measure phenomenon is referring to. Could the author explain the intuition and motivation clearly?


- Sec. 3.3 and Sec. 3.4 tried to draw a connection between successor features and FB. The introduction of the connection is informal. Is there any formal connection between those two families of methods?

- The reasoning that the expected variance of the returns vanished in Proposition 4.2 is handwavy. Specifically, from Eq. 26 to Eq. 27, the author claimed that the eigenvalues of the covariance matrix will converge to zero. This claim relies on the assumption that the observation has only finitely many effective dimensions. It is not clear whether this assumption will hold in practice.

- The behavioral task prior in Sec 4.2 was introduced using a heuristic. Will this prior prevent the issue mentioned in Proposition 4.1?

- In addition, the dataset of subtrajectories $D_{\text{sub}}$ is the power set of the offline dataset $D$, which contains infinite amount of subtrajectories even though $D$ is finite. If we sample all possible subtrajectories from $D_{\text{sub}}$, will it results in $p_{\text{data}}$ converging to $p_{\text{uniform}}$ asymptotically? Practically, how does the algorithm behave when $N_{\tau}$ goes up?

- In Appendix E.1., the paper shows ablation studies against other task sampling strategies based on a heuristic. Comparing the proposed method to some task sampling strategies from prior work (e.g., [1][2]) would strengthen the results.

[1] Frans, K., Park, S., Abbeel, P. and Levine, S., 2024. Unsupervised zero-shot reinforcement learning via functional reward encodings. arXiv preprint arXiv:2402.17135.

[2] Zheng, C., Park, S., Levine, S. and Eysenbach, B., 2025. Intention-Conditioned Flow Occupancy Models. arXiv preprint arXiv:2506.08902.

**Limitations:**

The paper concludes without discussing limitations. There are a couple of limitations that are worth mentioning:

- Is the linear reward assumption strong?

- Does the theoretical analysis in Proposition 4.1 rely on some assumptions?

- Connection with the prior method that infers (task) latent variables from the dataset.

**Strengths And Weaknesses:**

**Strengths**

- The paper highlights an under-explored but important design choice in offline zero-shot RL, namely how task (latent) vectors are sampled during training. This is a meaningful question because much prior work emphasizes representation learning while taking the reward/task distribution largely for granted.

- The proposed modification is simple and broadly applicable. In principle, BTD can be plugged into several existing pipelines without changing the underlying representation learning objective, which makes the idea easy to understand and potentially easy to adopt.

- The empirical section is reasonably broad in terms of the number of baselines. The method is evaluated on several representation learning approaches, including FB, and the paper also includes ablations on task dimension, interpolation with uniform sampling, and the number of GMM components. Those results provide evidence for claims in the paper.

**Weaknesses**

- Some theoretical results are provided informally, making it difficult to justify whether those claims are correct or not. See the question part for details.

- There is a gap between the theory and the algorithm. The theory is phrased in terms of the behavioral space $\Psi$, feature occupancies of policies, and variance over the full policy space $\Pi$. However, the actual method uses empirical discounted feature occupancies of random subtrajectories from the offline dataset. It is not clear whether the proposed method prevents the issue in Proposition 4.1.

-  The experiments focus on locomotion tasks from the ExORL benchmarks. It is not clear whether the conclusions from this paper generalize to some manipulation tasks or tasks that use visual inputs.

---

> ### Author Rebuttal · Authors · 2026-03-30
>
> We thank the reviewer for the feedback. We appreciate the recognition of our method’s simplicity, broad applicability, and thorough empirical evaluation. Below, we address questions on concentration of measure, connections to successor features, theoretical assumptions, the behavioral task prior, subtrajectory sampling, and comparisons to prior task-sampling methods.
>
> # W3. Generalization to visual inputs and manipulation tasks:
> We include two visual environments, Cheetah RGB and Walker RGB from Deepmind Control Suite and datasets from Exorl, as well as a manipulation task: Cube-single from ogbench. Due to space constraints, we do not report the table of results here and instead refer the reviewer to our response to Reviewer ysQX Q1 for the complete comparison.
>
> # Q1. Concentration of measure:
> The intuition we intended to convey is purely geometric. In high dimensions, a random direction (uniformly sampled tasks) is nearly orthogonal to any fixed low-dimensional subspace (the actual achievable behaviors). Because the reward $R_z(\psi) = \psi^\top z$ is the projection of the behavior onto the task direction, orthogonality results in near-zero rewards. This wastes the agent’s capacity on meaningless tasks. We agree that introducing the concentration of measure at that stage is premature and confusing. We will revise the introduction to provide a clearer explanation.
>
> # Q2. Formal relation between FB and SFs:
> Yes, there is a formal connection between FB and SFs. In fact, FB relies on a stronger condition that encompasses SFs: any FB representation $(F, B)$ satisfying the successor measure decomposition implicitly defines a set of successor features where $\psi(s,a,z) = F(s,a,z)$ and the underlying state features are $\varphi(s) = (\mathbb{E}_{\rho}BB^{\top})^{-1}B(s)$ (Section 3.4). Please refer to [3] Section 4.
>
> [3] Touati, A., Rapin, J., & Ollivier, Y. (2022). Does zero-shot reinforcement learning exist?. arXiv:2209.14935.
> # Q3. Formal proof and the underlying assumption:
> While mathematical edge cases exist (e.g., isotropic noise scaling with $d$), our assumption is based on the fixed complexity of real-world MDPs. Since the MDP dynamics are fixed, the intrinsic rank of feature occupancies is bounded regardless of the representation dimension $d$. As $d \to \infty$, the total signal is distributed over increasingly many redundant dimensions, forcing the eigenvalues associated with newer dimensions to decay and the variance to vanish. We will clarify this in the revision.
> # Q4. Behavioral task prior:
> While we do not provide a formal proof for the behavioral prior in Section 4.2, the intuition for why it prevents the vanishing variance is related to the structure of the task space. In Proposition 4.1, uniform sampling spreads the tasks across all $d$ dimensions, most of which are irrelevant. In contrast, the behavioral task prior samples from the empirical distribution of feature occupancies. This concentrates task vectors in the low-rank subspace where the behavioral signal actually lies.
> Importantly, assuming the offline dataset is sufficiently diverse, this prior does not hinder generalization to novel tasks. Instead, it ensures the task distribution covers the full span of achievable behaviors while ignoring the non-functional dimensions that cause the variance to vanish.
>
>
>
> # Q5. Dataset of subtrajectories:
> If dataset $D$ has $n$ trajectories, each with $m$ transitions (finite), the set $D_{sub}$ has $nm(m+1)/2$ subtrajectories (finite as well). Sampling all subtrajectories from $D_{sub}$ covers the dataset feature occupancy distribution. We are not trying to cover the sphere, but rather the empirical behavioral space of the dataset.
> Below is an ablation study on FB (BTD) with varying $N_\tau$:
>
> $N_\tau$|10|100|1k|10k|100k|1m
> -|-|-|-|-|-|-
> Cheetah|56±12|67±9|350±14|598±11|610±13|603±10
> Walker|57±7|32±10|388±12|628±8|623±14|631±9
> Quadruped|113±15|165±6|405±11|723±13|713±7|718±12
>
> Analysis:
> Small $N_\tau$ gives poor task distribution estimates and low performance. Increasing $N_\tau$ improves both, with performance saturating around 10k, indicating sufficient coverage and diminishing returns beyond.
>
> # Q6. Comparison to [1] and [2]:
> We provide a comparison with FRE [1] in the table below. However, [2] is not a suitable baseline for the “strictly” zero-shot RL setting addressed in this paper. Unlike our method, [2] requires a fine-tuning phase on reward-labeled datasets to estimate a Q-function and extract a policy.
> |Sampling Strategy|Cheetah|Walker|Quadruped|
> |-|-|-|-|
> |Uniform|464 ± 60|560 ± 42|634 ± 124|
> |Full trajectory|482 ± 37|488 ± 11|421 ± 30|
> |Subtrajectory|512 ± 24|579 ± 25|678 ± 43|
> |FRE [1]|319 ± 18|389 ± 31|453 ± 40|
> |**BTD-sampling (ours)** |**579 ± 25**|**611 ± 23**|**704 ± 64**|
>
> # Conclusion:
> We thank the reviewer again for these valuable suggestions. This feedback, the suggested limitations and the resulting clarifications will help us strengthen the final version of the paper.

---

> > ### Author Rebuttal · Reviewer_xNGi · 2026-04-01
> >
> > Thanks for answering my questions. Most of my questions have been addressed. The current rating can positively support your work, and I will update the presentation score accordingly. One remaining question:
> >
> > If I understand correctly, the subtrajectory does not necessarily contain consecutive transitions. So is the size of the set $D_{\text{sub}}$ being $nm(m+1)/2$ or $n2^m$? If it is the latter, $N_{\tau}$ will grow exponentially with the size of $D$.
> >
> > It would be good to include some discussions about the growth of $N_{\tau}$ and its effect. Meanwhile, I encourage the author to incorporate new experimental results and clarifications.

---

> > > ### Author Response · Authors · 2026-04-02
> > >
> > > Thank you very much for the follow-up. We want to address two points of confusion, both caused by imprecise wording on our part:
> > >
> > > First, **Subtrajectories are contiguous sequences of consecutive transitions**, not arbitrary subsets. The size of $D_\text{sub}$ is therefore $ |D_\text{sub}| = n m (m+1) / 2 $, which is polynomial in $m$. This is a natural choice, since contiguous subtrajectories correspond to coherent behavioral segments. We believe the confusion arose from our sentence:
> > >
> > >    > "we note $D_\text{sub}$ the set of all possible subtrajectories of varying lengths extracted from $D$"
> > >
> > >    which we will replace with:
> > >
> > >    > "all possible contiguous subtrajectories of consecutive transitions".
> > >
> > > Second, **$N_\tau$ is simply the number of subtrajectories we randomly sample from $D_\text{sub}$ to fit the GMM.** It is a fixed hyperparameter and does not grow with $|D_\text{sub}|$ or the dataset size.
> > >
> > > We will clarify all of this in the final version, along with an extended discussion on the effect of $N_\tau$ on performance and the additional experimental results.
> > >
> > > We are truly grateful for your careful reading. Your questions have directly helped us identify and fix gaps in our paper.

---

### Decision · Program_Chairs · 2026-04-30

**Decision:**

Accept (regular)

**Comment:**

All reviewers think that that the paper studies a relevant well-motivated problem and proposes an interesting solution with good empirical validation. All concerns, mostly about clarity of the presentation, have been nicely addressed by the rebuttal. I am thus recommending acceptance and encourage the author to revise the manuscript following the reviewers' feedback.